# Eusocial Transition in Blattodea: Transposable Elements and Shifts of Gene Expression

**DOI:** 10.3390/genes13111948

**Published:** 2022-10-26

**Authors:** Juliette Berger, Frédéric Legendre, Kevin-Markus Zelosko, Mark C. Harrison, Philippe Grandcolas, Erich Bornberg-Bauer, Bertrand Fouks

**Affiliations:** 1Institut de Systématique, Évolution, Biodiversité (ISYEB), Muséum National d’Histoire Naturelle, CNRS, Sorbonne Université, EPHE, Université des Antilles, CP50, 57 rue Cuvier, 75005 Paris, France; 2Institute for Evolution and Biodiversity, Molecular Evolution and Bioinformatics, Westfälische Wilhelms-Universität, Hüfferstrasse 1, 48149 Münster, Germany; 3Department of Protein Evolution, Max Planck Institute for Developmental Biology, Max-Planck-Ring 5, 72076 Tübingen, Germany

**Keywords:** termite, cockroach, major evolutionary transition, comparative genomics, comparative transcriptomics, molecular evolution, adaptive evolution, transposons

## Abstract

(1) Unravelling the molecular basis underlying major evolutionary transitions can shed light on how complex phenotypes arise. The evolution of eusociality, a major evolutionary transition, has been demonstrated to be accompanied by enhanced gene regulation. Numerous pieces of evidence suggest the major impact of transposon insertion on gene regulation and its role in adaptive evolution. Transposons have been shown to be play a role in gene duplication involved in the eusocial transition in termites. However, evidence of the molecular basis underlying the eusocial transition in Blattodea remains scarce. Could transposons have facilitated the eusocial transition in termites through shifts of gene expression? (2) Using available cockroach and termite genomes and transcriptomes, we investigated if transposons insert more frequently in genes with differential expression in queens and workers and if those genes could be linked to specific functions essential for eusocial transition. (3) The insertion rate of transposons differs among differentially expressed genes and displays opposite trends between termites and cockroaches. The functions of termite transposon-rich queen- and worker-biased genes are related to reproduction and ageing and behaviour and gene expression, respectively. (4) Our study provides further evidence on the role of transposons in the evolution of eusociality, potentially through shifts in gene expression.

## 1. Introduction

Eusociality, one of the major evolutionary transitions [1], refers to a social organization defined by cooperative brood care, overlapping generations, and a division of labour into reproductive and non-reproductive castes [2]. Eusociality has appeared independently across different groups in the tree of life, but is mainly known in arthropods, particularly in hymenopterans (ants, bees, and wasps) and in termites, as well as in crustaceans (snapping shrimp). The earliest appearance of eusociality, in termites, is supposedly older than 150 Ma (reviewed in [3]). The emergence of eusociality through natural selection can be explained by kin selection theory [4] coupled with high relatedness enhanced by monogamy [5]. Eusocial transitions led to complex innovations, such as cooperative behaviour with task specialization, complex communication, and increased fertility and longevity (breaking the fecundity/longevity trade-off) [6]. These adaptations led to viewing eusocial colonies as a “super-organism”, similar to the evolution of multicellular organisms, where the reproductive caste is seen as the germline and the non-reproductive caste as the soma, specialising in specific tasks [7,8]. In addition, eusociality led to a change of population structure, reducing the effective population size and increasing the generation time effect, which impacts major factors of evolutionary processes [9,10,11]. Indeed, a reduced effective population size weakens selection strength and strengthens the effect of genetic drift [12]; a longer generation time leads to an increased mutation rate [10]. However, the molecular basis underlying such complex adaptations remains elusive.

Some of the most puzzling genomic innovations are triggered by Transposable Elements (TEs), mobile genetic elements that were once seen only as genomic parasites (reviewed in [13]). While most TEs reach fixation within genomes through genetic drift [14], TEs can promote the ability of populations to adapt to rapidly changing environments (reviewed in [15,16]). Indeed, TE insertions can significantly alter phenotypes, contrary to point mutations. There are five main different mechanisms through which TEs can drive adaptive evolution: domestication/exaptation, where TEs give rise to new proteins; the change of gene expression due to nearby TE insertion; gene duplication with TEs inserting transcribed genes into other genome locations; fast evolving genomic regions; chromosomal rearrangements after ectopic recombination, resulting in the pairing of similar TE copies present in different chromosome regions (reviewed in [15]). The different mechanisms by which TEs can drive adaptive evolution along with their involvement in adaptive radiation in bats, primates, and lizards [17,18,19] indicate the pivotal role of TEs in species diversification. This is further supported by the fluctuation of TE propagation over evolutionary time with bursts of TE propagation following environmental stress and TE horizontal transmission (reviewed in [16,20]). However, the link between the TE-mediated phenotypic innovations driven by a specific mechanism is still missing.

In social bees, a decrease in transposable element diversity and abundance correlates with social complexity. Such a decrease could act positively on reducing kin conflict, thereby increasing relatedness among nest mates and, hence, indirect fitness [21]. Similarly, ancestral reconstruction of TE abundance and diversity in snapping shrimp indicates a moderate abundance of TEs predates and may have driven eusocial transition [22]. However, contrary to eusocial bees, eusocial snapping shrimp exhibit higher accumulation of TEs compared to non-eusocial species [22]. While the reduction of TE abundance could have favoured eusocial transition, eusocial-related demographic changes can lead to TE accumulation [22]. The reduction of TEs found in eusocial bees might be due to a trade-off between genomic diversity and integrity driven by recombination (diversity) and TE silencing (integrity) found in relation to increasing colony size in honey bees [23]. To further our understanding of the TE evolutionary dynamics and role in adaptive evolution during eusocial transition, we need to expand our investigation to more clades where eusociality has emerged.

Blattodea includes termites and cockroaches. All termites are eusocial, whereas cockroaches exhibit different social organizations, from solitary to subsocial ways of life [24]. Given this diversity and the latest advances in their phylogenetic relationships [25,26,27], this group is becoming a great model for comparative evolutionary analyses. Moreover, termites are insects with a major economic and ecological importance, but they remain understudied, especially when compared to Hymenoptera. Yet, they drastically differ from eusocial hymenopterans, being hemimetabolous (no metamorphosis to reach adult stage), and from snapping shrimp, as the reproductive division of labour is based on life stages, where workers are juveniles (nymphs). As eusocial shrimp, termites (but also cockroaches) exhibit high TE genomic content [28]; however, the TE evolutionary dynamics in these species remains largely unknown.

A study of three termite species and one gregarious cockroach species identified a genomic signature of eusociality in termites [28]. The presence of transposons in the flanking regions of expanding gene families could have facilitated the transition to eusociality. Indeed, a significant amount of insertions into regions adjacent to expanding gene families was observed in termites and absent in the gregarious cockroach, suggesting that these repeated element insertions would be shared among Isoptera. Gene duplication in eusocial hymenopterans has also been found to underlie eusocial transition [29,30,31]. Another major molecular basis underlying eusocial transition is an increase in gene regulatory potential, which could enable phenotypic plasticity [21,31,32]. A major impact of TE insertions near genes is a change of gene expression, through disruptions of regulatory regions and changes of the methylation state (reviewed in [16,33]). Hence, we hypothesised that TEs might have facilitated eusocial transition in termites through enhanced gene regulation.

In order to understand if TE-aided adaptive evolution could have led to shifts of gene regulation during eusocial transition in Blattodea, we categorized and annotated TE abundance and diversity in 2 cockroach and 4 termite genomes and determined gene expression bias in different life stages in those species. First, we investigated TE evolutionary dynamics in Blattodea. By mapping TEs onto the blattodean genomes, we investigated whether TEs insert more frequently in genes with differential gene expression between castes in termites and stages in cockroaches. More precisely, we focused on the queen (reproductive) and worker (sterile) castes in termites, and on nymphs (corresponding to the worker caste of termites) and female adults (corresponding to the queen caste of termites) in cockroaches. Furthermore, we explored if a high rate of TE insertions in genes with differential expression among castes in termites could be linked to functions important for eusocial adaptations. Our results indicated that TE accumulation within genes is driven mostly by selection and that a high rate of TE insertions in termite queen- and worker-biased genes seems to be linked to specific functions associated with eusocial adaptations in each caste.

## 2. Materials and Methods

### 2.1. Blattodea Genomes and Transcriptomes

To evaluate the role of TEs during the eusocial transition in Blattodea in connection with shifts of gene expression, the published genomes, as well as the transcriptomes of two life stages (worker, nymph and queen, adult female for termites and cockroaches, respectively) were retrieved for two cockroach species (*Blattella germanica*, *Periplaneta americana*) and four termite species (*Cryptotermes secundus*, *Reticulitermes speratus*, *Macrotermes natalensis*, *Zootermopsis nevadensis*; see Table 1).

### 2.2. Differential Gene Expression

To identify genes with biased expression in different life stages, two to six libraries of workers and queens of each termite species and four to six from nymphs and two from adult females in cockroaches were used (Appendix A). Those libraries were obtained from different tissues (whole body, whole body without gut, head) and from single or pooled individuals (Appendix A). The reads were mapped to the genomes using Hitsat2-2.1.0 [38] and were counted using Htseq-2.0.1 [39]. The differential expression analysis was performed with DESeq2 [40] between workers (majors and minors combined for *M. natalensis*) and queens in termites and between nymphs and adult females in cockroaches. We considered significant differential expression when the adjusted *p*-value was below 0.05 [40].

### 2.3. Transposable Element Annotations and Mapping onto Blattodea Genomes

To allow precise and complete detection of TEs for each species, a combination of ten tools was used: RepeatMasker (http://www.repeatmasker.org/), (accessed on 20 November 2021) RepeatModeler (http://www.repeatmasker.org/RepeatModeler/) (accessed on 20 November 2021), HelitronScanner [41], LTRpred [42], LTRharvest [42], mustv2 [43], SINE-scan [44], Sine-Finder [45], MiteFinderII [46], and tirvish [47] (http://genometools.org/tools/gttirvish.html) (accessed on 25 November 2021), implemented within the TransposonUltimate pipeline [48]. The de novo annotated TEs from each tool were combined and filtered using BEDtools (v. 2.27.1) [49] and a custom python script, removing duplicated or overlapping TEs (more than 20% overlap), as well as TEs of lengths above 10 kb. These filtered TEs were annotated to assign their class (RNA/DNA), subclass, and superfamily using the RFSB classifier from TransponUltimate [48] and as reference (training set) the TE database of insects from Petersen et al. (2019) [50]. The de novo TE database obtained for each species was then used to map TEs in each species’ genome using RepeatMasker (http://www.repeatmasker.org/) (accessed on 20 November 2021). The TEs obtained from RepeatMasker were further filtered to remove overlapping TEs, keeping the TE with the highest RepeatMasker score, and removing low-quality TEs, with a RepeatMasker score not greater than 1. Furthermore, Kimura distances accounting for the GC level were computed with a RepeatMasker script to evaluate the dynamics of the different TE superfamilies along Blattodea evolution. Lastly, to understand if TEs insert more frequently in genes with different expression levels among Blattodea species, we used the BEDtools closest tool categorizing for each gene the closest TE on the same DNA strand [49].

### 2.4. Gene Functional Annotations

To link the function of Differentially Expressed Genes (DEGs) between life stages where TEs insert and the social level of blattodean species, crowdGO [51] was used with the pre-trained crowdGOFull model [52,53,54,55] to annotate Gene Ontology terms [56] for each gene of each species. An enrichment of GO terms was tested with the GSEA test [57] with the “elim” algorithm implemented in the topGO package [58] in R [59]. Multiple datasets were used to categorize GO term enrichment. First, GO terms were analysed for enrichment of biased gene expression in worker/nymph and in queen/female adult in termites/cockroaches, separately. For these analyses, the gene score used was the negative log-transformed adjusted *p*-value of expression bias. Genes with non-computed adjusted *p*-values were discarded. Furthermore, the gene score was set to 1 for genes that displayed opposite biased expression of the life stage investigated for GO term enrichment. Secondly, GO term enrichments of genes with TEs inserted within them were performed separately for termites and cockroaches and for genes with biased expression in the different life stages, including genes with no bias for any life stage (adjusted *p*-value ≥ 0.05). For the enrichment test, the gene score used was the ratio of TE insertions within a gene kilobase. A gene was considered TE-rich if its score was greater than the third quartile of the dataset. Lastly, the significant GO terms found enriched in TE-rich DEGs were filtered out. GO terms enriched in TE-rich DEGs were removed if they were also found to be significant in enrichment analyses (1) of DEGs in the same life stage at both social levels (eusocial and gregarious), (2) of TE-rich DEGs in the same life stage, but in a different social level, and (3) of TE-rich DEGs of different life stages (including no bias) in the same social level. This filtering ensured that our list of significant enriched GO terms of TE-rich DEGs were specific to the expression level and social level and had a high TE insertion rate. REVIGO was used to reduce, cluster, and visualize significant GO terms, using the SimRel algorithm and clustering of 0.9 [60].

### 2.5. Statistical Analyses

All statistical analyses were performed in R (v4.2.1) [59], and visualisations were built with ggplot2 R package [61].

#### 2.5.1. TE Content and Dynamics in Blattodea

Because genome size is often correlated with TE content [22,50], we used Phylogenetically Independent Contrast, as implemented in the ape package [62], to evaluate if it stands true among blattodean species and if it could be attributed to specific TE superfamilies.

#### 2.5.2. TE Insertions, Biased Gene Expression, and Social Level

To investigate if TEs were inserted more frequently in genes with different expression levels among worker/nymph and queen/female adult depending on the social level of blattodean species (cockroaches: gregarious and termite: eusocial), the number of genes with a TE inserted or not was compared with regard to the gene expression bias and the social level, using phylogenetic generalized linear mixed models (PGLMMs) with a Poisson distribution and a Brownian motion covariance matrix from the brms package [63] (dependent variable: gene number, fixed factors: overlap with TE (yes or no) * social level (gregarious or eusocial) * gene expression bias (no bias or worker/nymph bias or queen/female adult bias), random factors: species + Brownian motion covariance matrix). Genes with no adjusted *p*-value for their expression bias were discarded from the analyses. Furthermore, we investigated if TEs present within genes insert more frequently within the exon or intron in relation to the social status of the species and the expression bias of the gene, and we built a PGLMM model as above (dependent variable: gene number, fixed factors: overlap with TE (intron or exon) * social level * gene expression bias, random factors: species + Brownian motion covariance matrix).

As the probability of TE insertions within genes is highly “co-”dependent on gene length, the difference of gene length (in kb) among social levels and different gene-biased expression were analysed with a GLMM with a Gamma distribution and species as a random factor implemented in the lme4 package [64] (dependent variable: gene length, fixed factors: social level * gene expression bias, random factors: species). As gene length differs significantly, the proportion of genes with a TE inserted within them between social level and gene expression bias was analysed with a GLMM with a binomial distribution and species and gene length as random factors to confirm the result of the PGLMM (dependent variable: gene overlap with TE (no = 0 or yes = 1), fixed factors: social level * gene expression bias, random factors: species + gene length).

In addition, to understand if TE superfamilies contributed differently to insertions within genes in cockroaches and termites, the number of TEs inserted within genes was compared in regard to the four main TE superfamilies and the social level, using phylogenetic generalized linear mixed models (PGLMMs) with a Poisson distribution and a Brownian motion covariance matrix from brms package [63] (dependent variable: TE number, fixed factors: overlap with gene (yes or no) * social level (gregarious or eusocial) * TE superfamily (*Tc-Mariner*, *hAT*, *Gypsy*, *LINE*), random factors: species + Brownian motion covariance matrix). The contribution of the different TE superfamilies to insertions within DEGs was also accounted for, by analysing the number of TEs within DEGs with a PGLMM model as above (dependent variable: TE number, fixed factors: social level (gregarious or eusocial) * TE superfamily (*Tc-Mariner*, *hAT*, *Gypsy*, *LINE*) * gene expression bias (no bias or worker/nymph-bias or queen/female adult-bias), random factors: species + Brownian motion covariance matrix).

All statistical models were simplified with a stepwise factor deletion and compared with each other, ensuring that the best model was kept.

## 3. Results

### 3.1. Differential Gene Expression

To investigate the role of TEs in shifts of gene expression during the eusocial transition in Blattodea, the analysis of the DEGs was performed on the entire transcriptomes of the 2 cockroach (gregarious) and 4 termite (eusocial) species. Genes were categorized as no bias, nymph-biased, or adult-female-biased in gregarious species and in eusocial species as no bias, worker-biased (corresponding to nymph developmental stage in cockroaches), or queen-biased (corresponding to adult female of cockroaches). Within the cockroaches *B. germanica* and *P. americana*, a total of 11,165 and 1814 genes, respectively, were detected as differentially expressed. The low amount of DEGs found in *P. americana* is likely the result of a methodological artefact. This impeded our comparative analyses as the results for gregarious species would mainly rely on *B. germanica*, since genes with no adjusted *p*-value for gene expression level, computed with DESeq2, were discarded; see above. Moreover, the differences in methodology to obtain transcriptomes for each species might bias the comparison of DEGs among species, especially when different tissue types are used, as gene expression is often tissue-specific. The proportion of genes highly expressed in nymph and adult females with a ≥2-fold change amounted to 916 (8.2%) and 849 (7.6%) in *B. germanica* and 31 (1.7%) and 6 (0.33%) in *P. americana*. In eusocial species, the amount of DEGs was relatively similar: *C. secundus* (13,761), *M. natalensis* (8340), *Z. nevadensis* (14,078), and *R. speratus* (13,874). The proportion of worker- and queen-biased genes was 344 (2.5%) and 729 (5.3%) in *C. secundus*, 1751 (21%) and 667 (8%) in *M. natalensis,* 549 (3.9%) and 816 (5.8%) in *Z. nevadensis*, and 555 (4%) and 860 (6.2%) in *R. speratus*.

### 3.2. TE Content and Dynamic in Blattodea

The TE landscape of blattodeans is largely dominated by Long Interspersed Nuclear Element (LINE) retrotransposons, followed by Tandem Inverted Repeat (TIR) DNA transposons, mainly the *Tc-Mariner* and *hAT* superfamilies, and by Long Terminal Repeat (LTR) retrotransposons, *Gypsy* (Figure 1A). Moreover, *Z. nevadensis* and *C. secundus* displayed higher proportions of *Tc-Mariner* DNA transposons and *Gypsy* retrotranspons, respectively, compared to other species (Figure 1A). The TE dynamics in the studied blattodean species, revealed by the sequence divergence of TE copies with their TE reference sequence, indicated a more recent TE expansions in cockroaches compared to termites (Figure 1B). In addition, all species displayed two major events of TE expansion in their genomes, while *P. americana* and *R. speratus* displayed one major and no clear event of TE expansion (Figure 1B).

Genome size varies greatly among blattodean species, ranging from 0.493 in *Z. nevadensis* to 3.374 Gb in *P. americana*. No correlation was found between genome size and TE content (r = 0.624, t = 1.382, df = 3, *p* = 0.261). Nevertheless, when restricting the correlation to termite species, genome size was correlated with TE content (r = 0.998, t = 14.55, df = 1, *p* < 0.05).

### 3.3. TE Insertions, TE Superfamily, Biased Gene Expression, and Sociality

The proportion of TEs inserted within genes is globally higher in cockroaches (82.1%) than termites (71.8%; Figure 2). When considering genes with biased or non-biased expression, the proportions of inserted TEs showed an opposite trend in cockroaches compared to termites. While in cockroaches, most TEs inserted within genes with no expression bias, followed by adult-female-biased genes, in termites, most TEs inserted within worker-biased genes (Table 2, Figure 2A).

Similarly, when comparing proportions of TEs inserted within exons and introns, opposite trends can be observed between cockroaches and termites (Table 3 and Figure 2B). While those results can be partly explained by gene length (Figure 3), as gene length differs significantly among genes with different expression bias (Table 4), the opposite trend of TE insertions within genes with different expression bias observed between termites and cockroaches remains true when accounting for gene length (Table 4). Hence, our results demonstrated that TEs insert more frequently within worker- and queen-biased genes in termites and within genes with no or female adult bias expression in cockroaches (Figure 2A). Furthermore, a higher number of genes had a TE inserted within an exon when their expression was worker-biased in termites, while in cockroaches, a higher number of female adult-biased genes displayed a TE inserted within an exon (Figure 2B).

TE superfamilies vary in their contribution to insertions within genes between cockroaches and termites. In both groups, retrotransposon *Gypsy* had the highest proportion of TE inserted within genes. Nevertheless, in eusocial species, a higher proportion of retrotransposon LINE and a lower proportion of DNA *hAT* were inserted within genes compared to gregarious species (Table 5 and Figure 4A). When accounting for differential gene expression, the insertion of TEs within DEGs also differed among TE superfamilies and between social level, while in cockroaches, the proportions of all TE superfamilies were similar among the different DEGs; in termites, DNA transposons (*Tc-Mariner* and *hAT*) were more frequently inserted within genes with biased expression compared to retrotransposons (*Gypsy* and LINE). More precisely, a higher proportion of DNA transposons inserted within queen-biased genes, while a higher proportion of retrotransposons inserted within worker-biased genes (Table 6 and Figure 4B). The different proportions of TEs, per TE superfamily, inserted within worker- and queen-biased genes observed only in eusocial species suggests a potential role of specific TE superfamilies during the evolution of termites.

### 3.4. Functions of DEGs Containing High Rate of TE Insertions

To understand if a high rate of TE insertions within DEGs could be linked to specific functions important for adaptive evolution in termites and cockroaches, the enrichment of GO terms of TE-rich DEGs (>3rd quartile) compared to other DEGs was calculated in eusocial and gregarious blattodean species. Our analyses revealed no function to be enriched in TE-rich DEGs in adult female cockroaches, while 16 GO terms were enriched in TE-rich DEGs in cockroach nymph-biased genes, 34 in termite queen-biased genes, and 54 in termite worker-biased genes. In termites, TE-rich queen-biased genes are related to metabolism, growth, and reproduction (Figure 5A and Appendix A), while TE-rich worker-biased genes are mostly related to development, metabolism, gene expression, and behaviour (Figure 5B and Appendix A). In cockroaches, TE-rich nymph-biased genes are mostly related to cell cycle, development, and metabolism (Figure 5C and Appendix A).

## 4. Discussion

Investigating the role of TEs in social evolution, our study revealed opposite trends of TE insertions within genes with biased expression in cockroaches and termites. Moreover, contrary to cockroaches, in termites, different TE superfamilies have more frequent insertions within genes according to their expression bias between castes. In cockroaches, TEs inserted more frequently, regardless of their superfamily, in genes with no bias in expression or in genes over-expressed in adult females, while nymph-biased genes showed a reduced number of TE insertion. In termites, on the other hand, TEs inserted more frequently, especially retrotransposons (*Gypsy* and LINE), in worker-biased genes (corresponding to nymph-biased genes in cockroaches). This trend could be explained by the differential selection strength on cockroach nymph-biased and termite worker-biased genes, as workers are not under direct selection, since they do not reproduce [67,68,69]. Hence, the reduced selection strength could explain the higher proportion of TE insertions in worker-biased genes in termites. This was confirmed by the highest propensity of TEs to insert within exons of worker-biased genes in termites compared to other DEGs and to cockroaches. In cockroaches, TEs inserted more often in the exons of adult-female-biased genes. However, changes in population structure associated with eusociality did not lead to higher TE insertions within genes in those species, suggesting that weaker selection and genetic drift may not have played a major role in TE insertion within genes. An alternate non-exclusive hypothesis, to explain such differences in TE insertion within genes, would rely on the “superorganism” view. Within the “superorganism” view, queen-biased genes correspond to the germline genes, where TE silencing is predominant [70,71], while worker-biased genes correspond to somatic cells, where the deleterious mutagenic impact of TEs should not affect fitness as much as in the germline [72]. In cockroaches, the negative mutagenic impact of TEs on fitness should be greater in nymph-biased genes than adult-female-biased genes, as deleterious mutations on nymph-biased genes should lead to a drastic reduction in fitness as nymphs are not reproductively active. Furthermore, the lack of a pattern of TE insertions of different TE superfamilies within DEGs in cockroaches and the clear pattern in termites of retrotransposons and DNA transposons being more frequent in worker-biased and in queen-biased genes, respectively, suggest the potential role of specific TE superfamilies during the evolution of termites. Indeed, retrotransposons inserting more often in worker-biased genes may favour their shifts in expression during eusocial transition, as has been previously suggested with LINE retrotransposons favouring gene family expansions during termite evolution [28]. DNA transposons may also play an adaptive role in eusocial transition by inserting more frequently in queen-biased genes. This result is in favour of the role of TEs in adaptive evolution during eusocial transition in Blattodea.

To understand if TE-aided adaptive evolution could have, at least partly, driven the eusocial transition, notably by facilitating shifts in gene regulation, an essential process in eusocial adaptation [21,23,31,32], not only the more frequent insertion of TEs in caste-specific genes, but the function of those genes need to be linked with adaptations associated with eusociality.

Our analyses of functional categories enriched among TE-rich genes revealed a large number of TEs inserted in queen-biased genes that could be essential for reproductive functions of queens. Indeed, several functional categories that were enriched in TE-rich queen-biased genes are indirectly (“regulation of growth”, “fat cell differentiation”, and “anatomical structure morphogenesis”) and directly (“multi-multicellular process”, “regulation of reproductive process”, and “meiotic nuclear division”) related to reproduction. Hence, numerous TE insertions within queen-biased genes may have facilitated a change in their expression to enhance egg production. Moreover, a high rate of TE insertions in queen-biased genes is linked to functions that could explain ageing adaptations in termite queens, such as “cellular respiration”, “insulin receptor signalling pathway”, and “fat cell differentiation”. Oxidative stress is one of the major impacts of ageing [6], and mutations in the cellular respiration pathway could impact the effect of oxidative stress [73]. Moreover, a relationship between ageing and insulin signalling has been established in several species [74], notably some termite species [75]. However, no clear trend of differential gene expression involved in the insulin pathway has been found across eusocial insects in relation to ageing and fecundity [6,76]. Nevertheless, TE-rich queen biased genes involved in fat cell differentiation could correspond to the adaptation to reproduction and ageing in termites, as evidence suggests the role of vitellogenesis, primarily produced in fat bodies [77], in ageing and fecundity adaptation in eusocial insects [6]. Therefore, the insertion of TEs within genes involved in those functions could have facilitated the adaptation in termite queens, leading to the absence of a trade-off in fecundity/longevity. Another function enriched in TE-rich queen-biased genes that could be linked to eusocial adaptation is immunity (“lymphocyte mediated immunity”, “B cell activation”, “regulation of leukocyte differentiation”, “negative regulation of immune system process”). Indeed, social adaptation is thought to lead to increased threat from parasites, due to the high density of closely related individuals, providing a rich environment for parasites [78]. Hence, enhanced regulation of immunity in termite queens may be of prime importance to modulate immune response at different colony life stages [79]. Nevertheless, the eusocial transition in termites is accompanied by a reduction of immune gene diversity and activity [80], which could explain high TE insertions in those genes that may not be active.

In worker-biased genes, enriched functions of TE-rich genes linked with the regulation of gene expression (“positive regulation of gene expression”, “gene expression”, “RNA metabolic process”, “response to endogenous stimulus”) and behaviour (“behaviour”, “adult locomotory behaviour”, “nervous system process”) could be related to eusocial adaptation. Indeed, behaviour-related genes are likely important for cooperative and diverse tasks undertaken by workers in a termite colony [24]. Furthermore, the phenotypic plasticity observed in eusocial insects has been hypothesized to have been favoured by enhanced gene regulation [21,31,32]. Hence, the high rate of TE insertion in worker-biased genes linked with gene expression could have facilitated the eusocial transition, enabling shifts and plastic gene expression. Furthermore, high TE insertions in worker-biased genes have been linked to processes involved in interaction with the host, which could be related to symbiotic relationships between termite protozoan to digest cellulose [24]. Finally, the high rate of TE insertions in nymph-biased genes in cockroaches seems mainly to be linked to development (“tube development”, “cell communication”, and “cell differentiation”) and metabolism (“lipid biosynthetic process”, “regulation of metabolic process”, “positive regulation of molecular function”, and “organophosphate metabolic process”) and may enhance the regulation of gene expression during this life stage [24].

The fixation of TEs is mainly driven by genetic drift [14]; however, the reduced effective population size caused by eusocial adaptation [11] does not seem to have led to higher TE insertions in termites compared to cockroaches. Moreover, the insertion of TEs seems to be more recent, as TE copies are less divergent from their reference TE, in cockroaches compared to termites. However, the generation time effect, leading to higher mutation rates, expected in eusocial insects [10], might also lead to higher mutations of TEs in termites. The generation time effect could explain greater sequence divergence among related TEs observed in termites compared to cockroaches. While TE superfamily content varies across species, no clear trend was found of different TE superfamilies among eusocial and gregarious blattodeans.

Our study lays the groundwork for studying the role of TEs in shifts of gene regulation during the transition to eusociality and warrants further investigations on the subject to precisely pinpoint the impact of TEs during eusocial evolution. However, our study suffered from some caveats: (1) the lack of manual curation of TEs ensuring a highly reliable TE database in Blattodea, (2) the lack of standardization in the transcriptomic methodology and the surprisingly low number of DEGs found in *P. americana*, rendering our subsequent analyses biased toward *B. germanica* results for the gregarious species, and (3) the lack of direct evidence of the TE insertions impact on gene expression among gregarious and eusocial Blattodea, as TE insertion in genes could also lead to exon disruption, exonisation, the creation of a new polyadenilation site, and alternative splicing (reviewed in [16]). In order to further our understanding, Blattodea genomes are necessary, especially species with various social levels. In our study, we only considered gregarious *vs.* eusocial blattodeans, while a great variety of social levels are present in cockroaches, notably with subsocial behaviour in cockroach species or distinct eusocial levels in termites, with lower and higher termites [81]. In addition, to directly categorize the role of TEs in gene expression shifts during the eusocial transition, functional studies are needed to unravel the impact of DEGs with TEs on eusocial adaptation. Furthermore, follow-up comparative genomic studies are needed to unravel how and on which ortholog genes TEs can lead to a change in gene expression between cockroaches and termites, notably through investigating methylation state or chromatin access, and comparing orthologs with the opposite direction of differential expression between cockroaches and termites. Another important factor to further our understanding would be to include transcriptomes of more life stages, as well as better-quality, tissue-specific, and standardized transcriptomes. To conclude, our study provides the first evidence of TE-aided eusocial adaptation in termites through possible changes of gene expression in different castes.

## Figures and Tables

**Figure 1 genes-13-01948-f001:**
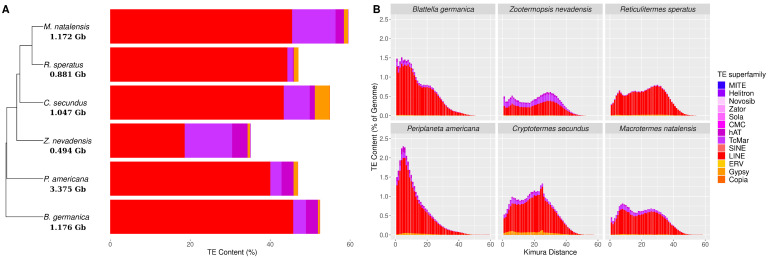
TE content and dynamics in Blattodea. (**A**) Phylogenetic tree of the 2 cockroach and 4 termite species with a histogram depicting TE abundance and diversity in each Blattodea species. The TE abundance and diversity was depicted as a percentage of TE content over genome size for each TE superfamily, and the genome size of each species is written below them in the phylogenetic tree. The phylogenetic tree was built with orthofinder (v. 2.5.4) [65] and visualized with ggtree [66]. (**B**) TE sequence divergence distribution per TE superfamily. The y-axis represents the percentage of TE content over genome size, and the x-axis corresponds to the Kimura sequence divergence (corrected for CpG) between single TE copies and reference consensus.

**Figure 2 genes-13-01948-f002:**
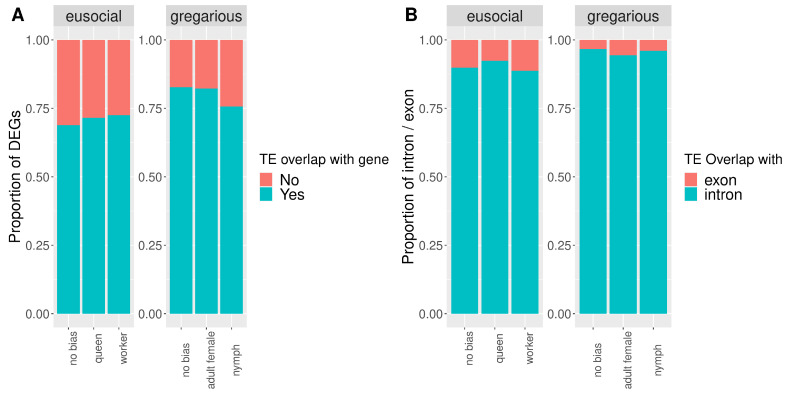
Proportion of differentially expressed genes with a TE insertion (**A**) and with a TE insertion within the exon or intron between cockroaches and termites (**B**). Genes are categorized as ‘no bias’ when they have no expression bias, queen-biased when their expression is higher in termite queens compared to workers, worker-biased when their expression is higher in termite workers compared to queens, female adult-biased when their expression is higher in cockroach female adults compared to nymphs, and nymph-biased when their expression is higher in cockroach nymphs compared to adult females.

**Figure 3 genes-13-01948-f003:**
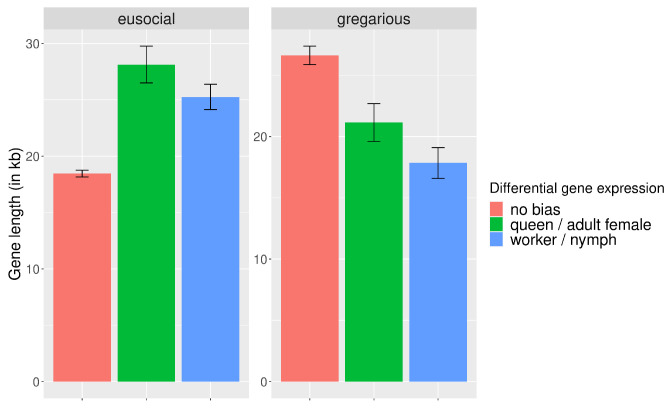
Length of differentially expressed genes between cockroaches and termites. The histogram depicts the average length of genes, and the error bars represent the 95% confidence interval.

**Figure 4 genes-13-01948-f004:**
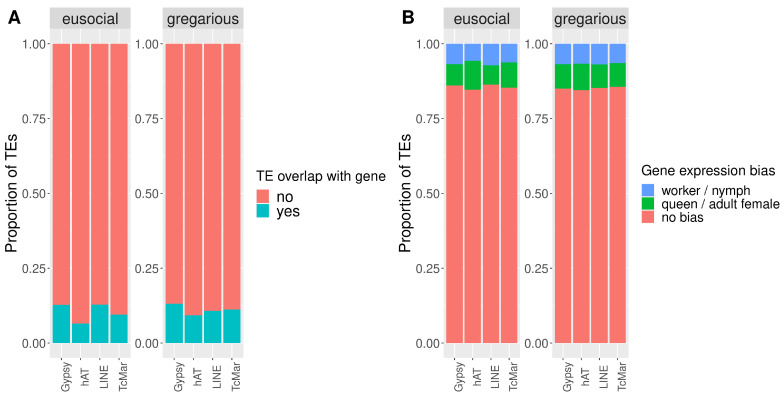
Proportions of TEs, for each TE superfamily, inserted within genes and DEGs between cockroaches and termites. (**A**) Proportions of TE insertion per TE superfamily between cockroaches and termites. (**B**) Proportion of TEs inserted within differentially expressed genes per TE superfamily between cockroaches and termites.

**Figure 5 genes-13-01948-f005:**
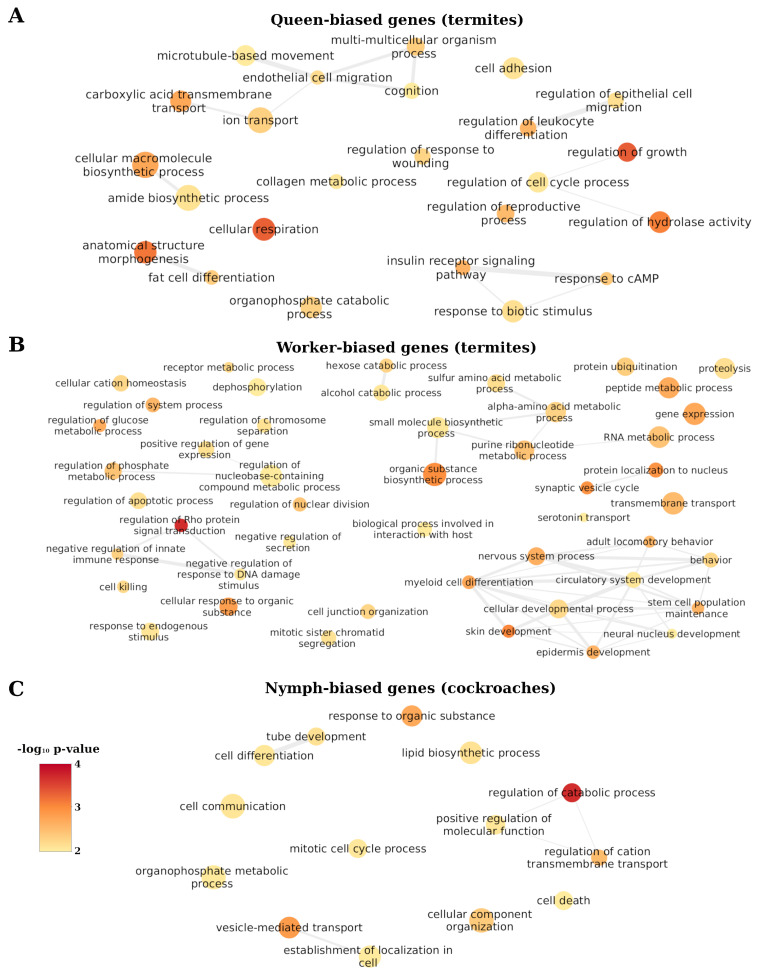
Functions enriched in TE–rich DEGs. (**A**) Functions significantly enriched in TE–rich termite queen-biased genes. (**B**) Functions significantly enriched in TE–rich termite worker–biased genes. (**C**) Functions significantly enriched in TE–rich cockroach nymph–biased genes. The colour of the circles represents their significance level and their size the number of annotated genes. REVIGO [60] and Cytoscape were used for visualization.

**Table 1 genes-13-01948-t001:** Accession details of genomes and transcriptomes of the different blattodean species and life stages.

Species	Genome Accession	Transcriptome: BioProject	Transcriptome: SRA Project	Reference
*B. germanica*	PRJNA427252	PRJNA382128	SRP104279	[28]
*P. americana*	PRJNA414776	PRJDB1997	DRP001319	[34]
*Z. nevadensis*	https://termitegenome.elsiklab.missouri.edu/ (accessed on 20 September 2021)	PRJNA203244	SRP022929	[35]
*C. secundus*	PRJNA381866	PRJNA382129	SRP104260	[28]
*R. speratus*	PRJDB2984	PRJDB3531	DRP002846	[36]
*M. natalensis*	http://gigadb.org/dataset/100057 (accessed on 30 September 2021)	PRJNA382034	SRP104234	[37]

**Table 2 genes-13-01948-t002:** Results of the best phylogenetic generalized linear mixed model explaining number of genes with TE insertions. The PGLMM was built with brms [63] with a Brownian Motion covariance matrix using 4 chains with 3000 warm-up rounds and 16,000 iterations. SL stands for Social Level, EB for Expression Bias, and G for overlap with Gene, and the interaction among fixed factors is denoted by “:”. To ease the visualization of the model’s results, the categorical terms were encoded as: gene overlap with a TE: 0 = no overlap and 1 = overlap; social level: 0 = gregarious and 1 = eusocial; gene expression bias: 0 = no bias, 1 = worker/nymph bias and 2 = queen/female adult bias.

	Estimate	Est.Error	Lower 95% CI	Upper 95% CI	Rhat	Bulk ESS	Tail ESS
Phylogeny	0.14	0.11	0.01	0.43	1	1459	2139
Species	0.79	0.62	0.04	2.37	1	1332	1581
			Fixed	Factors			
Intercept	6.29	1.19	3.92	8.80	1	1812	3271
SL	1.58	1.63	−1.86	4.93	1	1554	2254
EB	−1.45	0.04	−1.53	−1.37	1	972	2574
G	1.56	0.03	1.51	1.61	1	1526	3976
SL:EB	−0.17	0.05	−0.26	−0.08	1	964	2416
SL:G	−0.77	0.03	−0.83	−0.71	1	1523	4,105
EB:G	−0.17	0.05	−0.26	−0.08	1	954	2505
SL:EB:G	0.29	0.05	0.18	0.39	1	955	2,404

**Table 3 genes-13-01948-t003:** Results of the best phylogenetic generalized linear mixed model explaining the number of genes with TE insertions within introns or exons. The PGLMM was built with brms [63] with a Brownian motion covariance matrix using 4 chains with 4000 warm-up rounds and 40,000 iterations. SL stands for Social Level, EB for Expression Bias, and I/E for overlap with the Intron/Exon, and the interaction among fixed factors is denoted by “:”. To ease the visualization of the model results, the categorical terms were encoded as: feature overlap with a TE: 0 = intron and 1 = exon; social level: 0 = gregarious and 1 = eusocial; gene expression bias: 0 = no bias, 1 = worker/nymph bias, and 2 = queen/female adult bias.

	Estimate	Est.Error	Lower 95% CI	Upper 95% CI	Rhat	Bulk ESS	Tail ESS
Phylogeny	0.13	0.11	0.01	0.43	1	6168	8664
Species	0.79	0.63	0.04	2.4	1	7504	9870
			Fixed	Factors			
Intercept	7.86	1.18	5.46	10.3	1	11,012	16,057
SL	0.73	1.61	−2.55	4.04	1	9,234	14,634
EB	−1.64	0.02	−1.68	−1.59	1	8,681	18,225
I/E	−3.41	0.06	−3.54	−3.28	1	7,416	16,125
SL:EB	0.14	0.02	0.09	0.19	1	8,729	18,832
SL:I/E	1.24	0.07	1.11	1.38	1	7,401	15,883
EB:I/E	0.4	0.1	0.2	0.59	1	7,056	16,775
SL:EB:I/E	−0.51	0.11	−0.72	−0.3	1	7,080	16,599

**Table 4 genes-13-01948-t004:** Results of the best generalized linear mixed models explaining the gene length and the proportion of genes with TE insertions. The GLMM was built with lme4 [64] with, in both models, as a random factor the species, and gene length as a random factor in the model for the proportion of genes with TE insertions. SL stands for Social Level and EB for Expression Bias, and the interaction among fixed factor is denoted with “:”. To ease the visualization of the model results, the categorical terms were encoded as: social level: 0 = gregarious and 1 = eusocial; gene expression bias: 0 = no bias, 1 = worker/nymph bias, and 2 = queen/female adult bias.

		Estimate	Std. Error	t-Value	P
Gene Length	Intercept	36.6982	0.9905	37.051	$2\times 10^{-16}$
	EB	−1.3343	0.3445	−3.873	0.000108
	SL	−16.2092	1.3125	−12.35	$2\times 10^{-16}$
	EB:SL	3.3608	0.4022	8.356	$2\times 10^{-16}$
Proportion of genes with TEs	Intercept	2.68642	0.04745	56.617	$2\times 10^{-16}$
	EB	−0.14439	0.05478	−2.636	0.0084
	SL	−0.61644	0.03834	−16.079	$2\times 10^{-16}$
	EB:SL	0.14407	0.06046	2.383	0.0172

**Table 5 genes-13-01948-t005:** Results of the best phylogenetic generalized linear mixed model explaining the number of TE insertions within genes. The PGLMM was built with brms [63] with a Brownian Motion covariance matrix using 4 chains with 16,000 warm-up rounds and 40,000 iterations. SL stands for Social Level, TE for TE superfamily, and G for overlap with Gene, and the interaction among fixed factors is denoted with “:”. To ease the visualization of the model results, the categorical terms were encoded as: gene overlap with a TE: 0 = no overlap and 1 = overlap; social level: 0 = gregarious and 1 = eusocial; TE superfamily: *hAT* = 0, *Tc-Mariner* = 1, *Gypsy* = 2, LINE = 3.

	Estimate	Est.Error	Lower 95% CI	Upper 95% CI	Rhat	Bulk ESS	Tail ESS
Phylogeny	0.08	0.07	0.00	0.25	1.00	28,855	38,156
Species	0.49	0.41	0.02	1.57	1	30,591	42,338
			Fixed	Factors			
Intercept	10.03	0.69	8.58	11.50	1	52,704	42,160
SL	−0.64	0.94	−2.66	1.32	1	45,505	38,375
TE	1.53	0.00	1.53	1.53	1	99,078	77,999
G	−2.20	0.01	−2.21	−2.19	1	47,726	59,396
SL:TE	−0.19	0.00	−0.19	−0.19	1	101,434	75,867
SL:G	−0.49	0.01	−0.51	−0.47	1	48,328	60,044
TE:G	0.03	0.00	0.03	0.04	1	48,181	58,744
SL:TE:G	0.24	0.00	0.23	0.24	1	48,620	61,371

**Table 6 genes-13-01948-t006:** Results of the best phylogenetic generalized linear mixed model explaining the number of TE insertions within DEGs. The PGLMM was built with brms [63] with a Brownian Motion covariance matrix using 4 chains with 16,000 warm-up rounds and 40,000 iterations. SL stands for Social Level, EB for Expression Bias, and TE for TE superfamily, and the interaction among fixed factors is denoted by “:”. To ease the visualization of the model results, the categorical terms were encoded as: social level: 0 = gregarious and 1 = eusocial; TE superfamily: *hAT* = 0, *Tc-Mariner* = 1, *Gypsy* = 2, LINE = 3; gene expression bias: 0 = no bias, 1 = worker/nymph bias, and 2 = queen/female adult bias.

	Estimate	Est.Error	Lower 95% CI	Upper 95% CI	Rhat	Bulk ESS	Tail ESS
Phylogeny	0.07	0.06	0.00	0.24	1	28,259	33,354
Species	0.44	0.39	0.02	1.49	1	30,078	42,160
			Fixed	Factors			
Intercept	7.02	0.65	5.65	8.37	1	55,462	40,816
SL	−0.71	0.90	−2.59	1.12	1	47,606	34,390
EB	−1.54	0.02	−1.58	−1.51	1	53,309	64,186
TE	1.58	0.00	1.58	1.59	1	79,553	72,340
SL:EB	0.13	0.02	0.08	0.17	1	52,867	63,556
SL:TE	−0.03	0.00	−0.04	−0.02	1	78,169	72,885
EB:TE	−0.02	0.01	−0.03	−0.01	1	53,397	65,234
SL:EB:TE	−0.08	0.01	−0.10	0.07	1	53,218	62,503

## Data Availability

The de novo TE database of the 6 Blattodea species is freely available: http://dx.doi.org/10.17632/r7bwd37vcv.1, https://data.mendeley.com/datasets/r7bwd37vcv/1 (accessed on 20 November 2021).

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
