# Peer review of "Eusocial Transition in Blattodea: Transposable Elements and Shifts of Gene Expression"

_genes, 2022, doi:10.3390/genes13111948_

Round 1
Reviewer 1 Report
The work is very interesting, dealing with a complex topic, providingvaluable information about transposable elements. The different topics
are well described as well as the results correctly discussed. I suggest
that in Figure 1 the column indicating TE diversity is ordered according
to frequency.
Author Response
We would like to thank the reviewer for such positive comments. Figure 1 has been modified according to the reviewer’s suggestion, with now the genome content for each TE superfamily ordered according to frequency.
Reviewer 2 Report
The paper aims to find evidence for the molecular basis underlying the eusocial transition using cockroach and termite species as a model. Two concepts are linked together to uncover this evolutionary transition - 1) the transition to eusociality is accompanied by enhanced gene regulation, and 2) gene regulation can be influenced by transposon insertions. The manuscript is well-written and scientifically sound. Even though it is quite difficult to disentangle the effect of TE insertion on gene expression from the genetic effects themselves this paper provides interesting arguments for the field.
1) Line 128. The detected TEs are often used to demonstrate transposition events. In this part, I feel that TE is annotated de novo. Please fix.
2) Section differential gene expression in Results. It is not always clear which groups of comparison exactly are meant (gregarious vs eusocial or queens vs workers). It would be better to clarify this in this section.
3) LTR is Long Terminal Repeats, not Long Tandem Repeat. Please fix.
4) Line 228. I would recommend using terminology more carefully. Fixed TEs usually mean that TE insertion has been fixed during evolution. Most of these insertions have little or no effects on genome function and host fitness. That is why they become fixed. As far as I understood authors analyse recent insertions with little sequence divergence from the canonical sequence (it was not mentioned clearly in the text by the way). I would say there is not enough evidence to claim this insertion as fixed. But maybe I misunderstood something.
5) Figure 1. How phylogenetic tree was built and what exactly its branches show?
6) I am concerned about insertion into exons. At the first glance, I would say it completely suppresses the function of a gene. How do these insertions correlate with gene expression?
7) Figure 4. Presented GO terms seem to be redundant. I would suggest filtering to the top 10 according to gene ratio (DEG/all genes in category) and p-value.
Author Response
1) Line 128. The detected TEs are often used to demonstrate transposition events. In this part, I feel that TE is annotated de novo. Please fix.
> We changed it according to your suggestion, see line 130.
2) Section differential gene expression in Results. It is not always clear which groups of comparison exactly are meant (gregarious vs eusocial or queens vs workers). It would be better to clarify this in this section.
> We clarified the comparisons in this section, see lines 223-226.
3) LTR is Long Terminal Repeats, not Long Tandem Repeat. Please fix.
> Thanks for noticing this clerical mistake, changed line 245 and 464.
4) Line 228. I would recommend using terminology more carefully. Fixed TEs usually mean that TE insertion has been fixed during evolution. Most of these insertions have little or no effects on genome function and host fitness. That is why they become fixed. As far as I understood authors analyse recent insertions with little sequence divergence from the canonical sequence (it was not mentioned clearly in the text by the way). I would say there is not enough evidence to claim this insertion as fixed. But maybe I misunderstood something.
> Indeed, you are right we used the wrong terminology, we now have changed it lines 11, 249-250, 251, 252, 334, 339, 340, 341, and 413. For the analyses, we used the filtered list of TEs provided by RepeatMasker, including all TE sequences and therefore a large proportion of TEs with little sequence divergence from the canonical sequence. Hence, you are right we mostly analysed recent insertions.
5) Figure 1. How phylogenetic tree was built and what exactly its branches show?
> The phylogenetic tree was computed using Orthofinder and visualized with ggtree, we implemented it within the figure caption. The branches shows the branch length obtained from all gene tree inferences (STAG: https://github.com/davidemms/STAG). A clerical mistake led to wrong visualisation of branch lengths, it is now fixed, see Figure 1.
6) I am concerned about insertion into exons. At the first glance, I would say it completely suppresses the function of a gene. How do these insertions correlate with gene expression?
> Insertions into exons may result in diverse effects on a gene with several of them having an impact on gene expression: exonisation (modification of pre-exisiting exon), exon disruption (leading to gene silencing), creation of a new transcriptional start site (with TE inserting at the beginning of the first exon), and the creation of a new polyadelination site (with TE inserting at the end of the last exon) (reviewed in Pimpinelli and Piacentini 2020 Functional Ecology vol: 34 (2), 428-441).
7) Figure 4. Presented GO terms seem to be redundant. I would suggest filtering to the top 10 according to gene ratio (DEG/all genes in category) and p-value.
> We reduced redundant GO terms by adapting clustering coefficient, see new figure 5. We also include the full list of significant GO terms as a supplementary table, see lines 301, 303, and 304.
Reviewer 3 Report
This manuscript investigates the role of transposable elements in the transition to eusociality in termites and cockroaches. The authors combine genomic and RNAseq data to determine the potential role of TEs in the evolution of differential gene expression among castes and life stages of termites and their non-social relatives, cockroaches. The authors found that the abundance of TEs differ among genes that are differentially expressed in termites and that this enrichment seems to affect more strongly genes that could be involved in the evolution of eusociality. Although the present study does not prove a role of TEs, it adds to the literature that suggests that TEs may play a role in the transition from non-social to eusocial behavior. Overall, this is a well conducted study and the manuscript is clearly written. However, I have some comments that need to be addressed
The nature of the data and possible biases need to be discussed. Section 2.2 and table 1 should provide more information on the origin of the RNASeq data. I looked at a few of them on NCBI and it seems that not all datasets were obtained in similar ways. The RNASeq was obtained from degutted individuals in some species, while in other species pooled workers were analyzed. Also, I am wondering what is the effect of using whole individuals on RNASeq analyses. The potential biases (if any) related to different experimental design must be discussed.
Throughout the manuscript the authors refer to the fact that TEs insert preferentially (or not) in DEG. The abundance of TEs in a specific genomic region depends on insertion preference and on the rate of fixation (which depends on the effect of drift and selection). In their analysis the authors don’t present any evidence in favor of insertion bis and I would thus advise them to refrain from using wording like “preferential insertion” (e.g. Lines 177, 241, 250, 287).
The author did not find a correlation between genome size and TE content, which I found very surprising. It may be beyond the scope of this study but I wonder if the authors have any explanation for this observation. If not TEs, what accounts for the difference in genome size?
I would have liked to see more details about the differential abundance in TEs among DEGs. Are all TEs affected equally? Are some specific TEs enriched? I think it would be useful to know and it could affect the interpretation of the data. This information must be presented and discussed.
Although the writing is pretty good, in particular the introduction I found excellent, some sections of the discussion could be shortened. In particular I advise to remove the first paragraph (line 269-285) which only repeats what is already presented in the introduction and does not discuss the results of the study.
Minor comments:
Line 47: “resulting from a high number of localized TE insertions” needs tyo be replaced by a more specific statement on what TE mediated ectopic recombination is.
Line 77: a word is missing. I believe it should be “exhibit high TE genomic content”
Line 82: what is “they” referring to?
Line 96: It is “castes” not “caates”
Line 97: cockroaches is misspelled.
Line 221-222: LINE stands for Long Interspersed Nuclear Element
Line 223: LTR stands for Long Terminal Repeat
Line 374: “as TE copies are diverge less” should be replaced by “as TE copies are less divergent”
Author Response
The nature of the data and possible biases need to be discussed. Section 2.2 and table 1 should provide more information on the origin of the RNASeq data. I looked at a few of them on NCBI and it seems that not all datasets were obtained in similar ways. The RNASeq was obtained from degutted individuals in some species, while in other species pooled workers were analyzed. Also, I am wondering what is the effect of using whole individuals on RNASeq analyses. The potential biases (if any) related to different experimental design must be discussed.
> Indeed, this fact must be taken into account, we provide more details in the material and methods, see lines 116-118, and raise the possible shortcomings due to the disparity among the transcriptomes’ source and quality in the results, see lines 231-234, and discussion lines 425 and 443. The use of different source materials to obtain transcriptomes of different life-stages among species may lead to variation in categorizing genes as DEGs. Nevertheless, except from R. speratus, where transcriptome were obtained only from heads, the use of whole body with or without guts should not change the categorization of DEGs drastically.
Throughout the manuscript the authors refer to the fact that TEs insert preferentially (or not) in DEG. The abundance of TEs in a specific genomic region depends on insertion preference and on the rate of fixation (which depends on the effect of drift and selection). In their analysis the authors don’t present any evidence in favor of insertion bis and I would thus advise them to refrain from using wording like “preferential insertion” (e.g. Lines 177, 241, 250, 287).
> Thanks for letting us know about it, indeed it was not meant as an insertion preference, we fixed throughout the text as being more frequent insertions, see lines 9, 95, 142, 179, 189, 272, 326, 329, and 361.
The author did not find a correlation between genome size and TE content, which I found very surprising. It may be beyond the scope of this study but I wonder if the authors have any explanation for this observation. If not TEs, what accounts for the difference in genome size?
> The correlation between genome size and TE content was found only in termites, when including cockroach species this correlation was no longer true, mostly due to the great genome size of P. americana, in particular and cockroaches in general, but with relatively low TE content compared to other species. The reason behind this lack of correlation in cockroaches remains puzzling for us and we do not have any explanation for it, especially since P. americana genome is of the same quality as B. germanica genome.
I would have liked to see more details about the differential abundance in TEs among DEGs. Are all TEs affected equally? Are some specific TEs enriched? I think it would be useful to know and it could affect the interpretation of the data. This information must be presented and discussed.
> Right, thank your for this suggestion. We have now, implemented further analyses about the proportions of the TE insertions per TE superfamily within genes and DEGs. While, we found no pattern in cockroaches, in termites retrotransposons were more frequently inserted within worker-biased genes and DNA transposons within queen-biased genes. This result demonstrates that different TE superfamilies contributed differently in insertions within genes depending on gene expression bias, suggesting the different role of TE superfamilies during the transition to eusociality.
Although the writing is pretty good, in particular the introduction I found excellent, some sections of the discussion could be shortened. In particular I advise to remove the first paragraph (line 269-285) which only repeats what is already presented in the introduction and does not discuss the results of the study.
> The first paragraph of the Discussion section has been removed.
Minor comments:
Line 47: “resulting from a high number of localized TE insertions” needs tyo be replaced by a more specific statement on what TE mediated ectopic recombination is.
> Clarification has been made providing more details about TE-mediated ectopic recombination, see line 46.
Line 77: a word is missing. I believe it should be “exhibit high TE genomic content”
> Done line 76.
Line 82: what is “they” referring to?
> This has been changed, see lines 81-82.
Line 96: It is “castes” not “caates”
> Done line 96.
Line 97: cockroaches is misspelled.
> Done line 96.
Line 221-222: LINE stands for Long Interspersed Nuclear Element
> Done line 243 and 464.
Line 223: LTR stands for Long Terminal Repeat
> Done line 245 and 464.
Line 374: “as TE copies are diverge less” should be replaced by “as TE copies are less divergent”
> Done line 414.